# Analysis of mental and physical disorders associated with COVID-19 in online health forums: a natural language processing study

Rashmi Patel ,[1,2] Fabrizio Smeraldi ,[3,4] Maryam Abdollahyan,[5] Jessica Irving ,[1] Conrad Bessant [3,4]

RP and FS contributed equally.

¹Department of Psychological Medicine, Institute of Psychiatry Psychology and Neuroscience, King's College London, London, UK
²South London and Maudsley NHS Foundation Trust, London, UK
³Queen Mary University of London, London, UK
⁴The Alan Turing Institute, London, UK
⁵Barts Cancer Institute, London, UK

**Correspondence to**
Dr Rashmi Patel;
rashmi.patel@kcl.ac.uk

## ABSTRACT

**Objectives** Online health forums provide rich and untapped real-time data on population health. Through novel data extraction and natural language processing (NLP) techniques, we characterise the evolution of mental and physical health concerns relating to the COVID-19 pandemic among online health forum users.

**Setting and design** We obtained data from three leading online health forums: HealthBoards, Inspire and HealthUnlocked, from the period 1 January 2020 to 31 May 2020. Using NLP, we analysed the content of posts related to COVID-19.

**Primary outcome measures** (1) Proportion of forum posts containing COVID-19 keywords; (2) proportion of forum users making their very first post about COVID-19; (3) proportion of COVID-19-related posts containing content related to physical and mental health comorbidities.

**Results** Data from 739 434 posts created by 53 134 unique users were analysed. A total of 35 581 posts (4.8%) contained a COVID-19 keyword. Posts discussing COVID-19 and related comorbid disorders spiked in early March to mid-March around the time of global implementation of lockdowns prompting a large number of users to post on online health forums for the first time. Over a quarter of COVID-19-related thread titles mentioned a physical or mental health comorbidity.

**Conclusions** We demonstrate that it is feasible to characterise the content of online health forum user posts regarding COVID-19 and measure changes over time. The pandemic and corresponding public response has had a significant impact on posters' queries regarding mental health. Social media data sources such as online health forums can be harnessed to strengthen population-level mental health surveillance.

## INTRODUCTION

Measures to tackle the COVID-19 pandemic have resulted in unprecedented societal restrictions worldwide. The mental health impacts of these measures and accompanying socioeconomic stressors are likely to be extensive; identifying and quantifying these impacts are now an urgent priority.[1]

### Strengths and limitations of this study

► This study uses a novel approach to examine the content of a large, unstructured online health forum dataset using natural language processing.

► Online health forums provide a unique resource of real-world data that represent the lived experiences of people with mental and physical health disorders.

► It is not yet possible to establish COVID-19 status or whether concerned posters had pre-existing mental or physical health issues, had recovered or became unwell for the first time during the study period.

► Online health forums are help-seeking forums, which could contribute towards self-selection bias.

For example, social distancing restrictions make it harder to maintain regular contact between individuals and their friends and family as well as health and social care professionals. Furthermore, the psychological and emotional burden of the pandemic (and its consequences) may increase risk of relapse or worsen existing mental health disorders. Conversely, mental disorders can increase susceptibility to infections.[2 3]

Real-world data from online resources may be extracted using natural language processing (NLP) techniques to provide automated, population-level health surveillance. These methods can be used to rapidly ascertain discussion related to COVID-19 and associated symptoms and comorbidities. NLP has previously been used to identify medically relevant information from web pages and analyse extracted text.[4 5] Applying these techniques to real-world data sources such as social media and online forums may be used to supplement active data collection from participants in prospective observational research. Recent studies have applied this approach to Twitter, Facebook and Reddit data to forecast the emergence of depression

and post-traumatic stress disorder,[6] predict depression in the general population,[7] identify mothers at risk of post-partum depression[8] and investigate suicidal ideation.[9]

While social media platforms such as Twitter, Facebook and Reddit are commonly used, other internet resources such as online health forums have so far been neglected. Online health forums are enriched for health information and receive millions of posts each year, therefore providing untapped reservoirs of healthcare data at population level. Users of online health forums may post information or questions related to a particular health problem and discuss these in written threads with other forum users. Some users are 'peer experts' characterised as people with expertise in a particular area of health through lived experience. They may frequently respond to other users' questions and provide practical insights into managing a particular healthcare problem.[10] The typical demographic characteristics of online health forums include a greater proportion of female (78.4%) than male users with 18-year-old to 34-year-old users representing the most frequent age group.[11]

In a recent proof-of-concept study, we demonstrated that online health forums can be extracted to detect health discussion trends that correlate with real-life events.[12] Here, we use the same technology to analyse online health forum data discussing mental and physical health problems associated with the COVID-19 pandemic. We use NLP techniques to extract data from online health forum posts related to the COVID-19 pandemic, references to specific comorbid illnesses, and their direct and indirect impacts on mental or physical health.

## METHODS
### Study design and setting
We obtained data from online health forums using NLP. Online forums are discussion websites hosted on the internet where people hold conversations in the form of posted messages. A single conversation is called a thread. Threads are chains of posts identified within a forum by a title and an individual URL. Clicking on the thread title opens the thread which contains one or more posts which may be from the same user who started the thread (ie, the original poster) or different users who have replied within the thread. In this study, we analysed text data in thread titles and in individual posts within a thread. We analysed posts written in English only. Depending on the forum's settings, users can be anonymous or have to register with the forum to post messages, with most users opting not to use personally identifiable information to register their account. Registration may not be required for read-only access. Most forums recommend that users do not use personally identifiable information when posting. Online health forums specifically cover health topics and offer peer support for various health conditions.

We collected data from three major online health forums posted from 1 January 2020 to 31 May 2020: HealthBoards (www.healthboards.com), Inspire (www.

inspire.com) and HealthUnlocked (www.healthunlocked.com). These forums were chosen on the basis that they have global user coverage, include subforums on several aspects of healthcare, have a large user base contributing to regular activity on the forum and are feasible to extract information from using NLP.

HealthBoards was founded in California, USA, in 1997 and offers patient to patient health support. Inspire, founded in 2005, is a US healthcare social network managing online support groups for patients and caregivers. HealthUnlocked is a British online health forum launched in 2011 with a similar offering to HealthBoards and Inspire. Registration and participation in all three forums are free of charge to users.

### Analysis using NLP
#### Extracting and matching keywords
We extracted the keywords in thread titles and post content using lemmatisation. For flexibility and efficiency, search terms in posts and thread titles were matched using regular expressions that accounted for both inflection and common spelling variants. Online supplemental table 1 provides a key to the regular expression parameters used in the study. Matching was case-insensitive and limited to whole words in the post content and thread title; when matching thread URLs, parts containing words were considered. To prevent spurious matches, words shorter than four letters (eg, ICU) were considered valid matches only if they were delimited by non-word characters.

#### Definition of search terms
To investigate the potential impact of COVID-19 on users posting in online health forums, we classified threads and posts using keywords related to the COVID-19 pandemic in conjunction with various groups of case-insensitive keywords relating to (1) medical treatment in an intensive care unit or (2) physical symptoms as a direct consequence of COVID-19 infection or (3) mental health symptoms as a consequence of measures in response to the pandemic.

Search terms used to identify whether a thread or post was related to COVID-19 were 'COVID-19', 'COVID-19', 'coronavirus', 'corona', 'sars-cov-2', 'sars-2', 'shielding', 'pandemic*', 'vulnerable', 'quarantine', 'lockdown', 'distancing', 'isolation', 'isolating' where * indicates a wildcard search term.

Table 1 provides the final keywords used to search posts within COVID-19-related threads; the Python coded search terms are provided in online supplemental tables 2 and 3. We tested the specificity of keywords by searching for matches occurring before 1 January 2020. For threads, these were matches in the title and URL, while for posts, these were matches in the entire text (see Extracting and matching keywords above). Term incidence and excluded keywords are provided in online supplemental table 4.

**Table 1** Physical health, mental health and intensive care keywords used to search threads and posts

| | |
|---|---|
| Physical COVID-19 symptoms | chest pain, smell, taste, dry cough, anosmia, breath* |
| Mental health symptoms | worried, worry, worrying, worries, anxious, anxiety, feel(s) low, feeling low, depression, depressed, low mood |
| Intensive care terms | itu, icu, intensive care, intubation, intubated, ventilated, ventilator, c-pap, ecmo, membrane oxygenation |

### Data pre-processing

Data obtained from different online health forums come in various formats. We standardised and normalised the data before analysing them. This included normalisation of Unicode strings and whitespace characters, standardisation of date and time, and standardisation of location through the GeoNames.org database.

### Analysis of COVID-19 threads to identify changes in COVID-19-related user activity and physical and mental health associations over time

We identified the users contributing to a COVID-19-related thread in a given week. We then retrieved all the other posts made by the same authors in the previous, same and subsequent calendar weeks. We scanned such posts for physical symptom, mental health symptom or intensive care keywords as defined in table 1 and recorded whether each of these topics was mentioned by the author during the time window. We performed this analysis to establish variations in the prevalence of concerns relating to physical symptom, mental health symptom and intensive care keywords over the course of the pandemic during 2020. Weekly counts were measured each Sunday for the previous week.

### Analysis of thread titles

We inspected thread titles to identify how many mentioned a comorbidity in the title. We searched for terms related to autoimmune disorders, mental disorders or worry, cancer, cardiovascular problems or stroke, and diabetes as listed in table 2; the Python coded search terms are provided in online supplemental table 5.

### Analysis of first-time posters in a COVID-19-related thread

We analysed the first ever post published by a user to determine the proportion of first-time posters who started out by contributing to a COVID-19-related thread. We performed this analysis to determine the degree to which new users were motivated to make their first post in relation to the COVID-19 pandemic and how this varied over time during 2020.

### Implementation and computation

All descriptive analyses were performed using bespoke software written in Python. An outline of the coding approach employed is included in the online supplemental material.

### Data sharing

We consulted and adhered to internet research guidelines from the Association of Internet Researchers[13] and the British Psychological Society (BPS)[14] to inform study development.

Given licensing and privacy issues, it is not possible to publicly release the aggregate dataset generated from the three online health forums investigated. However, we welcome collaboration with other researchers and healthcare policy makers. Anyone interested in accessing the aggregate data and data analysis code should contact the guarantor (f.smeraldi@qmul.ac.uk).

### Patient and public involvement

Patients and the public were not involved in the design or conduct of the study.

**Table 2** Keywords applied to threads containing COVID-19 search terms to investigate physical and mental health diagnoses potentially associated with the COVID-19 pandemic

| Comorbid condition | Search terms |
|---|---|
| Cardiovascular/stroke | heart, infarct*, bypass, stent, coronary, ablation, atrial fibrillation, arrhythmia, aortic, cardio*, blockers, cardiomyopathy, statin(s), pressure, valve, pacemaker, stroke, ischaemia, ischaemic, anticoag*, anticoagulants, xarelto, apaxiban, rivaroxiban, dabigatran |
| Cancer | cancer, chemo, chemotherapy, tumours, melanoma, leukaemia, radiation, radiotherapy |
| Respiratory diseases | asthma, asthmatic, copd |
| Mental disorders | anxiety, anxious, depression, depressive, psychosis, psychotic, bipolar, schizophrenia, schizoaffective, ocd, ptsd |
| Autoimmune diseases | crohns, psoriasis, immunosuppress(ant), lupus, multiple sclerosis, ms, auto-immune |
| Diabetes | diabetes, mellitus, insulin, humira, remicade, metformin |

*A wildcard search.

**Table 3** Number of users and posts retrieved from selected online health forums from 1 January 2020 to 31 May 2020

| Website | Number of users* | Number of posts | Number (%) of posts mentioning COVID-19 |
|---|---|---|---|
| HealthUnlocked (www.healthunlocked.com) | 47 999 | 718 103 | 34 657 (4.8) |
| HealthBoards (www.healthboards.com) | 601 | 3477 | 127 (3.7) |
| Inspire (www.inspire.com) | 4534 | 17 854 | 797 (4.5) |
| Total | 53 134 | 739 434 | 35 581 (4.8) |

*A user is defined as anyone who posted within the data collection period.

**Table 4** Number and proportion of COVID-19-related threads and posts contained within COVID-19-related threads that mention a comorbidity

| Comorbidity | Number of threads, n (%)* | Number of posts contained within threads, n (%)† |
|---|---|---|
| No condition of interest | 2340 (70.0) | 34 201 (76.2) |
| Autoimmune disorders | 254 (7.6) | 2739 (6.1) |
| Cancer | 223 (6.7) | 2114 (4.7) |
| Respiratory | 150 (4.5) | 1905 (4.2) |
| Cardiovascular/stroke | 133 (4.0) | 1583 (3.5) |
| Diabetes | 45 (1.3) | 295 (0.7) |
| Mental disorders | 215 (6.4) | 2243 (5.0) |

Percentages do not add up to 100% because some threads contained mentions of more than one comorbidity.
*Percentage is calculated as number of mentions of comorbidities/total threads (N=3342).
†Percentage is calculated as number of posts/total posts (N=44 894).

## RESULTS

### Related posts and active threads

HealthUnlocked was the most frequently used forum accounting for 97% of overall posts and 97% of posts mentioning COVID-19 in the thread title or post content during the study period (table 3).

Weekly post count for HealthUnlocked peaked in mid-March. Post count for Inspire declined sharply in the last 2 weeks of March. Post count for HealthBoards declined slowly across the entirety of the observation period (online supplemental figure 1).

Across all three forums, there were a total of 3342 threads containing a COVID-19 keyword within the thread title or URL. These contained a total of 44 894 posts during the study period (1 January 2020 to 31 May 2020). A total of 35 581 posts (whether in COVID-19-related threads or otherwise) contained a COVID-19 keyword during the study period. The proportion of posts containing COVID-19 keywords increased rapidly across all forums in early March (online supplemental figures 2 and 3), corresponding with the WHO's declaration of COVID-19 as a pandemic on 11 March 2020. The smaller online forums (Inspire and HealthBoards) had a greater peak in percentage of total posts containing COVID-19 keywords. The total number of posts containing COVID-19 search terms declined from mid-April onwards.

For quite a long period, most posts about COVID-19 (over 90% at the beginning of the observation period and remaining above 50% until the week ending 29 March) were written by users who had not yet posted on the topic. By the end of the observation period, the percentage of weekly posts in COVID-19 threads written by new entrants to the discussion reduced to a still quite sizeable 30%. While many of these users may have posted before on the forum about other topics, online supplemental figure 4 presents the proportion of posters whose very first post to a forum appeared in a COVID-19-related thread. This figure peaked above 20% in the week ending 22 March. Considering that these forums have a very broad spectrum, this is a remarkably high fraction. It includes both new joiners and users who were previously silent members of the forums, possibly for a long time (so-called 'lurkers'), and who may have been spurred into a more active role by the pandemic.

### Thread title analysis

Over a quarter of COVID-19-related thread titles mentioned another condition of interest (table 4). After cancer and autoimmune diseases, mental health represented a major area of concern for online health forum users posting about COVID-19 comparable to respiratory and circulatory diseases (table 2). Around 0.5% of thread titles mentioned two or more comorbidities.

### User analysis

Posts in threads related to COVID-19 were analysed to determine the number of users contributing in each given week. For each active user, all posts in the previous, same and following calendar weeks were scanned irrespective of thread for mentions of physical symptom, mental health symptom or intensive care keywords. The number of active users mentioning each of these concerns peaked in the week ending 22 March and subsequently declined but still remained elevated above the January baseline. In particular, users discussing mental health outnumbered users mentioning the other topics (figure 1).

## DISCUSSION

Using a novel technique to analyse data from online health forums, we found a marked increase in posts

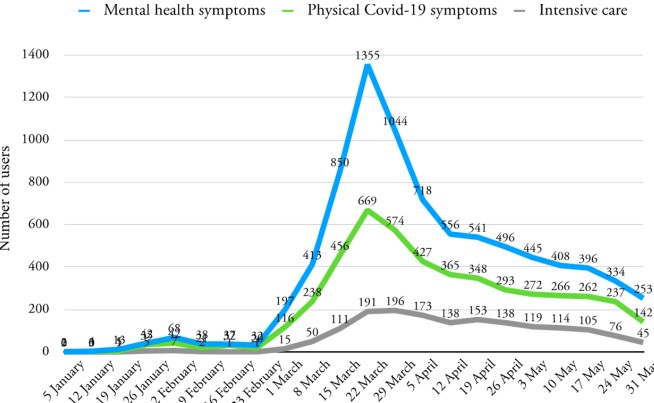

**Figure 1** Number of users making posts in threads related to COVID-19, which included physical symptoms, mental health symptoms or intensive care keywords.

related to COVID-19 across the observation period of 1 January 2020 to 31 May 2020. The frequency of these posts increased rapidly in early March 2020 corresponding with the WHO's declaration of COVID-19 as a pandemic.

During this period, we found mental health symptom keywords were most frequently mentioned by authors of COVID-19-related posts (either contextually or in separate messages), followed by physical symptoms and intensive care keywords, suggesting that the pandemic and public health response to it has had a significant impact on posters' concerns regarding mental health. The marked increase in mental health symptom-related posts in early March, when the WHO declared the COVID-19 pandemic, correlates with preliminary worldwide data that show increases in anxiety and depression in response to the outbreak.

The mental health impacts of COVID-19 and associated physical distancing restrictions are likely to be extensive and wide-reaching. There is a growing body of evidence supporting the neuropsychiatric effects of coronavirus infections.[15] Restrictions fuel socioeconomic stressors such as unemployment, loneliness and financial burden, which are all implicated in the development of mental ill health.[16] Increased rates of bereavement, newfound caring responsibilities and interruptions to education are likely to be particularly stressful to children and young adults.[17]

A preliminary survey of 3545 German respondents found evidence of substantial mental health burden from travel and physical distancing restrictions, including increased levels of stress, anxiety, depressive symptoms, sleep disturbance and irritability.[18] Worsening mental health has been confirmed in samples with both pre-pandemic and post-pandemic information for direct comparison: The Avon Longitudinal Study of Parents and Children (ALSPAC) study found probable anxiety disorder doubled compared with pre-pandemic sizes (26% vs 13%) and lower well-being, particularly in young people, women and those with pre-existing conditions.[19] The literature on social media mining for COVID-19 mental health-related trends is limited. A study analysing

sentiment evolution trends of four emotions across Twitter—fear, anger, sadness, joy—has been able to identify developing shared distress and topics of interest relating to those emotions.[20]

Our findings also suggest that mental and physical health concerns documented in online forum posts have levelled off following their peak in March 2020 but their rates continued to remain above pre-pandemic levels. The number of users active in COVID-19 threads who also wrote posts concerning mental health symptoms reduced from their peak in March of 1355 (per week) to 253 by the end of the observation period (compared with a mean number of 30 per week in January), suggesting that as time went on most users had begun to adjust to the consequences of the pandemic. Furthermore, the increased engagement of users on online health forums could be related to the restrictions on travel and in-person interaction during the pandemic, leading to a shift to social engagement through online media. Other NLP studies have also identified a similar trend. An analysis of 10 million Google searches within the USA found large shifts in mental health symptom searches linked to stay-at-home orders in the USA across the week commencing 16 March 2020.[21] Searches for topics related to anxiety, negative thoughts about oneself and the future, insomnia and suicidal ideation dramatically increased prior to stay-at-home orders, levelling off on the announcement of stay-at-home orders. These patterns were relatively unique to searches for mental health-related information and not physical conditions.

We used keywords related to intensive care treatment in NLP models reflecting the potential for COVID-19 to lead to serious illness requiring intensive care treatment. We found that threads containing content related to intensive care treatment increased in frequency during the pandemic. Follow-up data in more recent studies has highlighted the potential for COVID-19 to be associated with prolonged, multisystem involvement associated with significant disability,[22] with considerable reduction in functioning and quality of life among individuals who have received intensive care treatment.[23] A significant proportion of people who have received intensive care treatment for COVID-19 have post-intensive care syndrome characterised by physical and cognitive impairments and problems with mental health.[24] This could explain the increasing rates of threads related to intensive care treatment due to forum users' concerns of the potential need for intensive care treatment, raising questions about family members or friends currently receiving intensive care treatment, or in relation to problems faced after being discharged from the intensive care setting.

Over the entire period, on average 4% of first-time posters (over 20% in the peak period) made their very first contribution to the forum in a COVID-19-related thread. Furthermore, 77% of COVID-19 threads were started by users who had never posted about the topic before and chose to start out by creating their own thread. A certain degree of motivation is required to take someone to the

point of making that first post on a forum and also for starting a thread; our finding suggests that the pandemic is driving users to engage more actively in community forum services in times of uncertainty.

## Strengths and weaknesses

Online health forums are an important source of real-world, real-time, population-level data on people living through the COVID-19 pandemic. Online health forums also afford users anonymity to discuss aspects of their experience they might otherwise have been embarrassed or fearful to disclose in identifiable forms of social media. We have demonstrated that it is possible to automate information extraction from these posts using NLP, providing access to a rich reservoir of previously untapped real-world data from health-specific online resources.

Our approach was able to automatically extract data from a large sample of over 53 000 unique users at a fraction of the cost of previous approaches that have relied on social media individual participant recruitment and manual review of posts generating sample sizes in the low hundreds.[7] Some studies screened users on Twitter via depression symptom questionnaires and used their tweets to train depression onset classifiers.[6 25] Analogous approaches have been used with Facebook data.[8]

Our study has some limitations. At present, it is difficult to establish whether concerned posters have pre-existing mental or physical health issues, have experienced confirmed COVID-19 illness themselves, are recovered or have become unwell for the first time. Online health forums are help-seeking communities; this introduces self-selection bias in which individuals from disadvantaged backgrounds who do not have IT equipment/network connection to access online resources are under-represented and our results are therefore not generalisable to the entire population. Furthermore, as these forums have worldwide coverage, we cannot isolate trends to one geographic region. However, future work could use the location data (see Data pre-processing in Methods) to explore this avenue. We analysed data up to 31 May 2020 providing a relatively short period of follow-up following the onset of the COVID-19 pandemic. Future studies examining a longer period of follow-up would help to determine if the increase in activity and content related to COVID-19 and comorbid health problems persisted. As we analysed anonymised data, we were unable to investigate the demographic characteristics of the forum users in our study to determine whether they were representative of the wider population.

## CONCLUSIONS AND FUTURE RESEARCH

Publicly accessible sources of real-world data, such as online health forums analysed in this study, can strengthen population-level physical and mental health surveillance and provide a rapid and inexpensive means to inform public healthcare policy. We found that the majority of posts in online forum data related to COVID-19

concerned features related to mental health and that the peak in frequency of posts corresponded with the early phase of the pandemic, indicating the significant impact of COVID-19 on the mental health of susceptible populations.

As the pandemic evolves, further research using online forum data could improve our understanding of the long-term consequences of COVID-19 infection[26] and the longer-term socioeconomic consequences of travel and physical distancing restrictions that have been employed in many countries to manage viral transmission.[27 28] Analysis of real-world data, including social media and online health forums, could provide a useful insight into attitudes and perceptions towards novel therapeutics. This will be crucial to maximising continued uptake of effective preventative approaches such as mask-wearing, physical distancing, ventilation, hygiene measures and vaccines.

**Contributors** The study was conceived by RP and FS. Data extraction and statistical analysis were performed by FS, MA and CB. Reporting of findings were carried out by RP and JI. All authors (RP, FS, MA, JI and CB) contributed to study design, manuscript preparation and approved the final version. FS is a guarantor.

**Funding** RP has received support from a Medical Research Council (MRC) Health Data Research UK Fellowship (MR/S003118/1), a National Institute for Health Research (NIHR) Advanced Fellowship (NIHR301690) and a Starter Grant for Clinical Lecturers (SGL015/1020) supported by the Academy of Medical Sciences, The Wellcome Trust, MRC, British Heart Foundation, Arthritis Research UK, the Royal College of Physicians and Diabetes UK. FS and CB were funded by an Alan Turing Institute (ATI) Fellowship and by an EPSRC COVID-19 Rapid Response Impact Acceleration Fund. Computational resources were funded by a Microsoft Azure Sponsorship through the ATI.

**Competing interests** RP has received funds from Janssen, Induction Healthcare and Holmusk outside the current study.

**Patient consent for publication** Not applicable.

**Ethics approval** Data for this study were drawn from publicly available online health forums and extracted in aggregate form for secondary data analysis rather than at individual user level. The study was independently reviewed by the Queen Mary Ethics of Research Committee (QMERC) and received an exemption for ethical approval. The data were analysed using the computing infrastructure based at Queen Mary University of London (QMUL) which employs a two-layer security model to maintain data privacy. QMUL is registered as a data controller with the Information Commissioner's Office (ICO; registration number: Z5507327), which covers all research activities undertaken at the university.

**Provenance and peer review** Not commissioned; externally peer reviewed.

**Data availability statement** No data are available.

**ORCID iDs**

Rashmi Patel http://orcid.org/0000-0002-9259-8788

Fabrizio Smeraldi http://orcid.org/0000-0002-0057-8940

Jessica Irving http://orcid.org/0000-0002-2847-6508

Conrad Bessant http://orcid.org/0000-0002-7983-1020

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
