## [Reviewer comments · BMJ Open]

ARTICLE DETAILS

TITLE (PROVISIONAL)	Analysis of mental and physical disorders associated with COVID-19 in online health forums: a natural language processing study
AUTHORS	Patel, Rashmi; Smeraldi, Fabrizio; Abdollahyan, Maryam; Irving, Jessica; Bessant, Conrad

VERSION 1 – REVIEW

REVIEWER	Nakanishi, Nobuto Tokushima University Hospital, Emergency and Critical Care Medicine
REVIEW RETURNED	11-Sep-2021

GENERAL COMMENTS	Thank you for the opportunity to review this article. This article is well written. No revision is needed for the publication. Well-done.
---

REVIEWER	González-Seguel, Felipe Clinica Alemana de Santiago SA, Servicio Medicina Física y Rehabilitación
REVIEW RETURNED	13-Sep-2021

GENERAL COMMENTS	Manuscript: bmjopen-2021-056601 Title: Investigating mental and physical disorders associated with COVID-19 in online health forums First author: Rashmi Patel Reviewer: Felipe González-Seguel Reviewer Comments: GENERAL COMMENTS: First, thank you for the opportunity to review this work. The aim of this study was to characterise the evolution of mental and physical health concerns relating to the COVID-19 pandemic among online health forum users using novel data extraction and natural language processing techniques. This original work displays the current problems related to the pandemic from a little-explored point of view. SPECIFIC COMMENTS: 1. Abstract: “We obtained data from 739,434 posts by 53,134 unique users”. This sentence should be in the result sub-section of the abstract, because is part of the findings after to apply the methods.
---

2. Abstract: The results subsection still needs to answer the "primary outcome measures" that were mentioned in methods, for example, the proportions obtained.

3. Introduction: To further contextualize, I suggest adding the information related to the magnitude and representativeness of the use of health forums (posts and threads) in the general population. For example, what is the proportion of people who use this type of online forum in your country (or in other countries)? What is the type of people who do it (characteristics)? All this could condition the generalization of the data to the general population. Please, if this is relevant, add it to the discussion-limitations if these results represent a small part/type of the population.

4. Methods and Results were clear.

5. Discussion: I suggest that the authors add a paragraph that contrasts their results with clinical results on the mental and physical impact of COVID-19, especially in patients who required a stay in the ICU. Although there are many studies that could be contrasted with the results of this study, I suggest the followings: doi: 10.1186/s13256-020-02481-y, doi: 10.1038/s41572-020-0201-1, doi: 10.1136/bmjopen-2021-053610, doi: 10.1136/bmjopen-2020-048392, doi: 10.1016/j.eclinm.2021.101019, doi: 10.1186/s13613-021-00910-9, doi: 10.3390/jcm10173870.

6. Congratulations to the team of researchers.

REVIEWER	Jefferson, Laura University of York, Department of Health Sciences
REVIEW RETURNED	29-Sep-2021

GENERAL COMMENTS	Thank you for the opportunity to read this interesting research article that presents a relatively novel approach to studying trends in social commentary relating to mental and physical disorders during the COVID pandemic. The study presents an excellent example of how data of this kind may be used to inform healthcare and government policy, in a relatively rapid and low-cost way. While I recommend the paper for publication, there are some minor suggestions that I would make:  1. Inclusion of some methodological descriptor in the study title may aid readers sifting studies. 2. The discussion suggests rates of discussion around mental health have "levelled off" but it is perhaps worth noting here that they are still higher than pre-pandemic levels. 3. Would it be worth continuing this work over a longer timescale, to explore trends across subsequent waves of the pandemic? This is a relatively short timescale covered. 4. More of a comment really and something we have found in our own research using social media data to explore commentaries around GP wellbeing during the pandemic (yet to be published) - it seems that increasing engagement in online forums may be a symptom of the society's reduced ability to engage socially through other means during the pandemic. This highlights the growing need for this kind of research method.
---

VERSION 1 – AUTHOR RESPONSE

Reviewer: 1

Dr. Nobuto Nakanishi, Tokushima University Hospital

Comments to the Author:

Thank you for the opportunity to review this article. This article is well written. No revision is needed for the publication. Well-done.

/*Thank you for your supportive comments regarding our study.*/

Reviewer: 2

Mr. Felipe González-Seguel, Clinica Alemana de Santiago SA, Universidad del Desarrollo

Comments to the Author:

GENERAL COMMENTS:

First, thank you for the opportunity to review this work. The aim of this study was to characterise the evolution of mental and physical health concerns relating to the COVID-19 pandemic among online health forum users using novel data extraction and natural language processing techniques. This original work displays the current problems related to the pandemic from a little-explored point of view.

/*Thank you for your supportive comments regarding our study.*/

SPECIFIC COMMENTS:

1. Abstract: "We obtained data from 739,434 posts by 53,134 unique users". This sentence should be in the result sub-section of the abstract, because is part of the findings after to apply the methods.

/*We have updated the abstract to move this statement to the results section.*/

2. Abstract: The results subsection still needs to answer the "primary outcome measures" that were mentioned in methods, for example, the proportions obtained.

/*We have updated the abstract to add details of the primary outcome measures in the results section.*/

3. Introduction: To further contextualize, I suggest adding the information related to the magnitude and representativeness of the use of health forums (posts and threads) in the general population. For example, what is the proportion of people who use this type of online forum in your country (or in other countries)? What is the type of people who do it (characteristics)? All this could condition the generalization of the data to the general population. Please, if this is relevant, add it to the discussion-limitations if these results represent a small part/type of the population.

/*We have updated the introduction section to cite a study analysing the demographic characteristics of people who use online health forums (Page 4, Paragraph 3). We are unable to characterise demographic characteristics in our study as we analysed an anonymised dataset with no personal or demographic information on the users. We have stated this as a limitation in the discussion section (Page 12, Paragraph 2).*/

4. Methods and Results were clear.

/*Thank you.*/

5. Discussion: I suggest that the authors add a paragraph that contrasts their results with clinical results on the mental and physical impact of COVID-19, especially in patients who required a stay in the ICU. Although there are many studies that could be contrasted with the results of this study, I suggest the followings: doi: 10.1186/s13256-020-02481-y, doi: 10.1038/s41572-020-0201-1, doi: 10.1136/bmjopen-2021-053610, doi: 10.1136/bmjopen-2020-048392, doi: 10.1016/j.eclinm.2021.101019, doi: 10.1186/s13613-021-00910-9, doi: 10.3390/jcm10173870.

/*We have updated the discussion section to include further comparisons with the results of this study. (Page 11, Paragraph 2).*/

6. Congratulations to the team of researchers.

/*Thank you for your supportive comments.*/

Reviewer: 3

Dr. Laura Jefferson, University of York

Comments to the Author:

Thank you for the opportunity to read this interesting research article that presents a relatively novel approach to studying trends in social commentary relating to mental and physical disorders during the COVID pandemic. The study presents an excellent example of how data of this kind may be used to inform healthcare and government policy, in a relatively rapid and low-cost way.

*/*Thank you for your supportive comments.*/*

While I recommend the paper for publication, there are some minor suggestions that I would make:

1. Inclusion of some methodological descriptor in the study title may aid readers sifting studies.

*/*We have amended the article title to provide description of the methods and study setting.*/*

2. The discussion suggests rates of discussion around mental health have "levelled off" but it is perhaps worth noting here that they are still higher than pre-pandemic levels.

*/*We agree with this assessment and have updated the discussion to note this (Page 10, Paragraph 5).*/*

3. Would it be worth continuing this work over a longer timescale, to explore trends across subsequent waves of the pandemic? This is a relatively short timescale covered.

*/*We agree that this work would be worth continuing over a longer time period based on the relatively long natural history of chronic health disorders that could be associated with the COVID-19 pandemic. We hope to do this in future studies and hope that this publication will encourage other research teams to apply the same methods in other datasets. We have updated the discussion section to make this recommendation (Page 12, Paragraph 2).*/*

4. More of a comment really and something we have found in our own research using social media data to explore commentaries around GP wellbeing during the pandemic (yet to be published) - it seems that increasing engagement in online forums may be a symptom of the society's reduced ability to engage socially through other means during the pandemic. This highlights the growing need for this kind of research method.

*/*Thank you for your insights and this would make sense given the restrictions on travel and in-person interaction during the pandemic. We have updated the discussion section to note this (Page 10, Paragraph 5).*/*

Reviewer: 1

Competing interests of Reviewer: None

Reviewer: 2

Competing interests of Reviewer: None

Reviewer: 3

Competing interests of Reviewer: None declared.

VERSION 2 – REVIEW

REVIEWER	González-Seguel, Felipe Clínica Alemana de Santiago SA, Servicio Medicina Física y Rehabilitación
REVIEW RETURNED	01-Oct-2021
GENERAL COMMENTS	I thank the authors for answering my concerns. I wish you all the best for the publication.
REVIEWER	Jefferson, Laura University of York, Department of Health Sciences
REVIEW RETURNED	05-Oct-2021
GENERAL COMMENTS	The authors have addressed the comments well, no further comments.